# Cross-Modal Meta Consensus for Heterogeneous Federated Learning

## ABSTRACT

In the evolving landscape of federated learning (FL), the integration of multimodal data presents both unprecedented opportunities and significant challenges. Existing work falls short of meeting the growing demand for systems that can efficiently handle diverse tasks and modalities in rapidly changing environments. We propose a meta-learning strategy tailored for Multimodal Federated Learning (MFL) in a multitask setting, which harmonizes intra-modal and inter-modal feature spaces through the Cross-Modal Meta Consensus. This innovative approach enables seamless integration and transfer of knowledge across different data types, enhancing task personalization within modalities and facilitating effective cross-modality knowledge sharing. Additionally, we introduce Gradient Consistency-based Clustering for multimodal convergence, specifically designed to resolve conflicts at meta-initialization points arising from diverse modality distributions, supported by theoretical guarantees. Our approach, evaluated as $M^3Fed$ on five federated datasets, with at most four modalities and four downstream tasks, demonstrates strong performance across diverse data distributions, affirming its effectiveness in multimodal federated learning. The code is available at https://anonymous.4open.science/r/M3Fed-44DB.

## CCS CONCEPTS

• **Information systems** → **Multimedia information systems**; • **Computing methodologies** → **Cooperation and coordination**.

## KEYWORDS

Multimodal Federated Learning, Meta-learning

## 1 INTRODUCTION

The emergence of multimodal federated learning (MFL), a novel paradigm allowing multiple parties to collaboratively train models using clients' multimodal data without compromising privacy, has garnered considerable attention. MFL [6, 18, 37, 47] focuses on how large-scale distributed clients can collaborate to train multimodal-related models (such as multimodal fusion [34], cross-modal translation [58], multimodal knowledge bases [8], etc.) without sharing data, where each client can collect multimodal data from various types of sensors (such as images, videos, audio, text, time series data, etc.). Intuitively, federated systems trained with multimodal

Permission to make digital or hard copies of all or part of this work for personal or classroom use is granted without fee provided that copies are not made or distributed for profit or commercial advantage and that copies bear this notice and the full citation on the first page. Copyrights for components of this work owned by others than the author(s) must be honored. Abstracting with credit is permitted. To copy otherwise, or republish, to post on servers or to redistribute to lists, requires prior specific permission and/or a fee. Request permissions from permissions@acm.org.

MM'24, October 28 - November 1, 2024, Melbourne, Australia.
© 2024 Copyright held by the owner/author(s). Publication rights licensed to ACM.
ACM ISBN 978-1-4503-XXXX-X/18/06
https://doi.org/XXXXXXX.XXXXXXX

**Unpublished working draft. Not for distribution.**

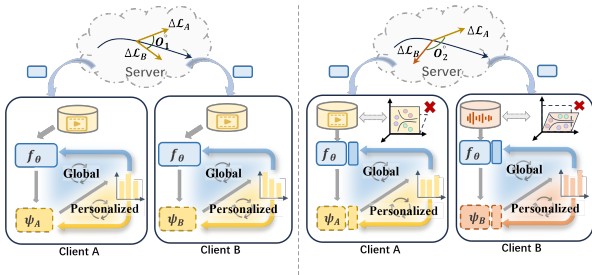

(a) Unimodal federated meta-learning     (b) Multimodal federated meta-learning

**Figure 1: Unimodal federated meta-learning *vs.* Multimodal federated meta-learning. ✖: Different modalities have inconsistent feature spaces. $O^\circ$: "The gradient optimization directions exhibit disparities, where $O_1^\circ$ is non-conflicting, while $O_2^\circ$ is conflicting."**

data are expected to be more robust and insightful compared to their single-modal counterparts.

There has been a growing body of work [10, 13, 51, 55] focusing on the task of multimodal federated learning (MFL). Recently, Yang et al. [49] propose the cross-modal federated human activity recognition where each client has only one type of modality. They disentangle the local model into modality-agnostic(shared across all clients) and modality-specific block (shared with the same modality). To further address the challenges posed by modality gaps, task gaps, domain shifts, and concept drifts among clients, Chen et al. [11, 12] propose a dynamic and multi-view graph structure to aggregate the different model block. This framework employs knowledge disentanglement to facilitate optimal information sharing among clients. It achieves this by transforming asymmetrical exchanges into symmetrical ones based on semantic knowledge, thereby significantly enhancing communication through a meticulously designed two-stage disentanglement process. Although the **disentangle-based** methods relieve the modality gaps and data heterogeneity, the process of disentangling models into modality-agnostic and modality-specific parts or into smaller blocks for different subsets of clients adds a layer of complexity in model architecture. This complexity can lead to increased computational demands during both the training and inference phases, possibly limiting the scalability of the approach to large-scale federated networks. Therefore, **disentangle-based** methods may not be as agile in adapting to entirely new tasks or rapidly changing data environments.

Meta-learning [14, 21, 23] stands out as an intuitive approach for Federated Learning since it is specifically designed to enable models to learn new tasks quickly and efficiently with minimal data. By facilitating rapid personalization and improving generalization, meta-learning enhances the effectiveness and efficiency of learning across decentralized datasets, aligning perfectly with the objectives

of federated learning. Unfortunately, this study focuses on applications in the single modality domain [9, 17, 22, 26, 39, 44, 48] with less consideration in practical settings of multi modal.

In the domain of multimodal federated learning, effectively aligning various modalities and tasks to enable meta-learning algorithms to utilize cross-modal information presents significant challenges. This process necessitates advanced methodologies to ensure that the integration of knowledge across tasks and modalities positively influences the learning mechanism. The key challenges include: ① As shown in Fig. 1a, existing federated learning approaches primarily focus on finding a cross-task meta-model within a single modality. This orientation neglects difficulties in facilitating meta-knowledge sharing that spans both intra-modal and inter-modal feature spaces among clients. Achieving meta-knowledge sharing across feature spaces of diverse modal tasks necessitates **intricate transformation mechanisms** that can reconcile the differences between modalities, ensuring that the federated learning process respects the unique characteristics of each modality while leveraging their complementary strengths. ② The goal of meta FL for single modality is to find an initial point shared between all clients which performs well after each user updates it with respect to its own loss function, potentially by performing a few steps of a gradient-based method. This endeavor becomes exponentially more challenging within the realm of multimodal federated learning, where client data spans diverse modalities—text, images, and audio—each characterized by unique feature spaces and statistical properties. As shown in Fig. 1b, there arise challenges of gradient conflicts when aggregating these multimodal clients models. The diversity requires the global models that **adapts to heterogerous multimodal cilents**, demanding advanced adaptation strategies.

In this paper, we propose a meta learning strategy for MFL under multitask setting ($M^3Fed$). For **intricate transformation mechanisms**, we introduce the concept of a Cross-Modal Meta Consensus Space, aiming to harmonize and integrate diverse modalities into a unified representation, facilitating seamless knowledge sharing and transfer across varied data types. Specifically, we propose a dual-level optimization architecture: the personalized optimization is dedicated to enabling task heterogeneity within the same modality, while the global optimization incorporates a consensus operator for facilitating the sharing of meta-knowledge across different modalities. For **adapting the meta-learning for FL of heterogerous multimodal cilents**, we propose a versatile Cross Modal Meta Aggregation scheme. We upload the meta learner and consensus operator to the server . We aggregate the meta learner based on gradient consistency-based clustering, leveraging similarity in optimization directions for aggregation across multimodal clients. This approach resolves gradient conflicts arising from distribution disparities, thereby attaining a more adaptable meta-model. For aggregating consensus operators, we employ a global consensus collaboration matrix to evaluate operator relevance, thereby facilitating more effective interaction among heterogeneous modal clients. Our contributions are summarized as follows.

1) We introduce a meta-learning strategy for multimodal federated learning that leverages the Cross-Modal Meta Consensus Space to enhance within-modality task adaptation and streamline cross-modality knowledge transfer.

2)We propose the gradient consistency for multimodal convergence addressing conflicts at meta-initialization points arising from varied modality distributions with theoretical guarantees.

3)We evaluate $M^3Fed$ on five federated datasets, with at most four modalities and four downstream tasks. The empirical results demonstrate the effectiveness of our method.

## 2 RELATED WORK

### 2.1 Multi-Modal Federated Learning

Compared to unimodal federated learning [45, 46, 59], multimodal federated learning (MFL) has received growing attention in recent years due to its potential to cover a wider range of practical application scenarios. In existing studies, two main configurations have been explored: homogeneous multimodal federated learning, in which each client has a complete modal dataset; and heterogeneous multimodal federated learning, in which there is missing modal data between clients, resulting in heterogeneity of modal distributions among different clients [29]. Considering that heterogeneous multimodal federated learning is closer to real-world complexity, this paper will focus on this configuration. In the research field of heterogeneous MFL, current approaches [11, 13, 49, 55] mainly adopt the strategy of submodule training, which facilitates knowledge sharing among clients by aggregating submodules that contain modal shared knowledge. For example, Yang et al.[49] propose a modality collaborative activity recognition network, which can collaboratively learn a global activity classifier shared across all clients and a modality-dependent private activity classifier based on modality-agnostic and modality-specific features respectively with the guide of an adversarial modality discriminator. Chen et al.[11] propose FedMSplit, which employs a dynamic graph structure to adaptively capture the relationships among different types of clients and then achieve correlated model training. To further address the challenges posed by modality gaps, Chen et al. [12] transform asymmetrical exchanges into symmetrical ones based on semantic knowledge, thereby significantly enhancing communication through a meticulously designed two-stage disentanglement process.

However, current approaches based on model separation or feature decoupling rely on complex model architectures containing multiple sub-modules designed for different data modalities, which not only increases the difficulty of deployment on resource-constrained devices, but also increases the communication and computational burden during joint learning. By introducing meta-learning, our approach enables a unified model to quickly recognize and adapt deep features and modalities when encountering different client data. This not only simplifies the model architecture and reduces the reliance on large amounts of data, but also enhances the model's adaptability to local data, leading to more efficient and flexible learning in heterogeneous MFL environments.

### 2.2 Federated Meta Learning

Federated meta learning [2, 35, 57] aims to train a model that is quickly adapted to new tasks with little training data, where clients serve as a variety of learning tasks. The seminal model-agnostic meta-learning (MAML) framework [19] has been intensively applied to this learning scenario. Some work [7, 43] has begun to

explore how to combine federated learning and meta-learning, leveraging the advantages of meta-learning to address issues such as personalization [17, 22, 44, 48] and accelerated convergence [9, 26, 39] in federated learning. For example, Jiang et al. [22] use a unified perspective on federated meta-learning to compare MAML and the first-order approximation approach. Fallah et al. [17] present Per-FedAvg, which learns an initial shared model, enabling rapid adaptation and personalization for each client. FedMeta [50] is a two-stage optimization with a controllable meta updating scheme after model aggregation. A federated meta-learning technique called MetaGater is proposed by Lin et al. [27]. It trains the channel gating and backbone network simultaneously. By utilizing model similarity across learning tasks on various nodes, MetaGater ensures the effective capture of relevant filters for speedy adaptation to new tasks, making it possible for resource-constrained applications to select subnets efficiently. Experimental findings validate the efficacy of MetaGater. Yang et al. [48] propose G-FML, which adaptively divides the clients into groups based on the similarity of their data distribution, and the personalized models are obtained with meta-learning within each group. While existing approaches have made some progress in personalization and rapid adaptation, these achievements have mainly focused on the domain of unimodal federated learning, while meta-learning research in multimodal federated learning environments is still rarely addressed. This paper proposes Cross-Modal Meta Consensus for Heterogeneous Federated Learning and provides an in-depth exploration of multimodal federated meta-learning, aiming to fill this research gap.

## 3 MODIFICATIONS

### 3.1 Problem Formulation

We posit the existence of a trustworthy server and $K$ clients operating within the framework of federated learning. Clients maintain a veil of secrecy between them, rigorously safeguarding personal privacy data. Each client, denoted as $i$, possesses an arbitrary-sized private local modal dataset $D_i = \{X_i, Y_i\}$, $i = 1, 2, ..., K$. In our investigation, we contemplate a heterogeneous federated learning arrangement where each client harbors its own autonomous task $\mathcal{T}$ alongside separate modal $\mathcal{M}$ data. These clients collaboratively, through identification and knowledge integration, train a global meta-learning model $F(\cdot; \theta) : \mathbb{R}^n \rightarrow \mathbb{R}^d$, where $\theta$ is the model parameters, $n$ and $d$ are the dimensions of the input data and extracted features of the input data, respectively. Building upon meta-learning framework, each user initializes from the meta-model $F$ and subsequently updates using the gradient descent of their own loss function. Therefore, the overall optimization objective of federated learning is as follows:

$$\min_\theta F := \frac{1}{K} \sum_{i=1}^{K} F_i(\theta - \eta \nabla F_i(\theta)), \qquad (1)$$

where $\eta$ is the stepsize. The advantage of this formula lies in its ability to preserve the advantages of federated learning (FL) while also capturing the differences between users' various tasks and modalities. Users can utilize the solution to this new problem as a starting point and perform slight updates based on their private data.

In this work, we define the function $F$ in Eq.1 as a basic feature meta learner $f_\theta$ and the meta-consensus subspace projection mechanism $G, T$, with detailed specifics to be elaborated in Section 3.2. By employing meta-learning within an Dual-level optimization framework, local clients facilitate the transfer of shared knowledge among modalities, thus constructing a cross-modal consensus feature space. To enhance the adaptability of the meta-model for aggregating heterogeneous modality models, the server selectively aggregates based on the similarity of gradient optimization directions (Section 3.3). We conduct theoretical analysis of gradient consistency-based clustering in Section 3.4.

### 3.2 Localized Training Via Dual-level Optimization

**Projection Metric.** The collection of all $t$-dimensional linear subspaces in a $D$-dimensional space $\mathbb{R}^D (0 < t \leq D)$ is referred to as the Grassmann manifold $\mathcal{G}(t, D)$, which represents smooth surfaces embedded in high-dimensional Euclidean space. Previous research [1, 24] has proposed that the distance between subspaces can be computed using geodesic distance. Following the prior article [38], we employ projection metric $\varrho^2$[20] as the similarity for subspace distance, defined as follows:

$$\varrho^2(V_A, V_B) = tr[(V_A - V_B)^\top (V_A - V_B)] = \|V_A - V_B\|_F^2 \qquad (2)$$

where $\|\cdot\|_F$ denotes the Frobenius norm; $V_A, V_B$ are the orthogonal projections of two subspaces $A, B$.

Inspired by the algorithm for subspace projection metric in Eq.2, we can utilize this formula to measure the distinctiveness of feature spaces. In our study, data from different clients exhibit disparities in both modality and task. Consequently, there exists significant inconsistency in the feature spaces extracted by the consensus meta-model. To address this issue, we propose a trainable subspace orthogonal projection operator. Through this operator, we are able to transform the fundamentally disparate feature spaces into a meta-consistent embedding space, thereby facilitating knowledge transfer among clients.

**Consensus Subspace Projection Mechanism.** The heterogeneity in client sensor configurations implies disparities in both modality and task within their data. This impedes the attainment of a globally consistent feature space during federated training. Traditional federated learning methods may suffer performance degradation when handling diverse feature spaces, as they struggle to extract complementary knowledge from other modality data. To address this issue, we propose a Consensus Subspace Projection Mechanism designed at the client-side. This operator learns how to project different base feature spaces onto specific consensus subspaces, facilitating knowledge propagation.

As shown in Fig.2, in our approach, there are two types of learnable operators for achieving consensus subspace projection transformation: personalized operator ($T$) and shared consensus operator ($G$). $T$ is designed for client-specific data to learn personalized knowledge information, whereas $G$ is shared and employed for inter-client knowledge transfer tasks. Specifically, samples $x$ from the client's private dataset $D_i$ is transformed into an $n$-dimensional vector through a shared meta learner $f_\theta$. Subsequently, personalized operators $T_i$ project the features into a consensus subspace, yielding $T_i \cdot f_\theta(x)$. Ultimately, this process culminates in obtaining

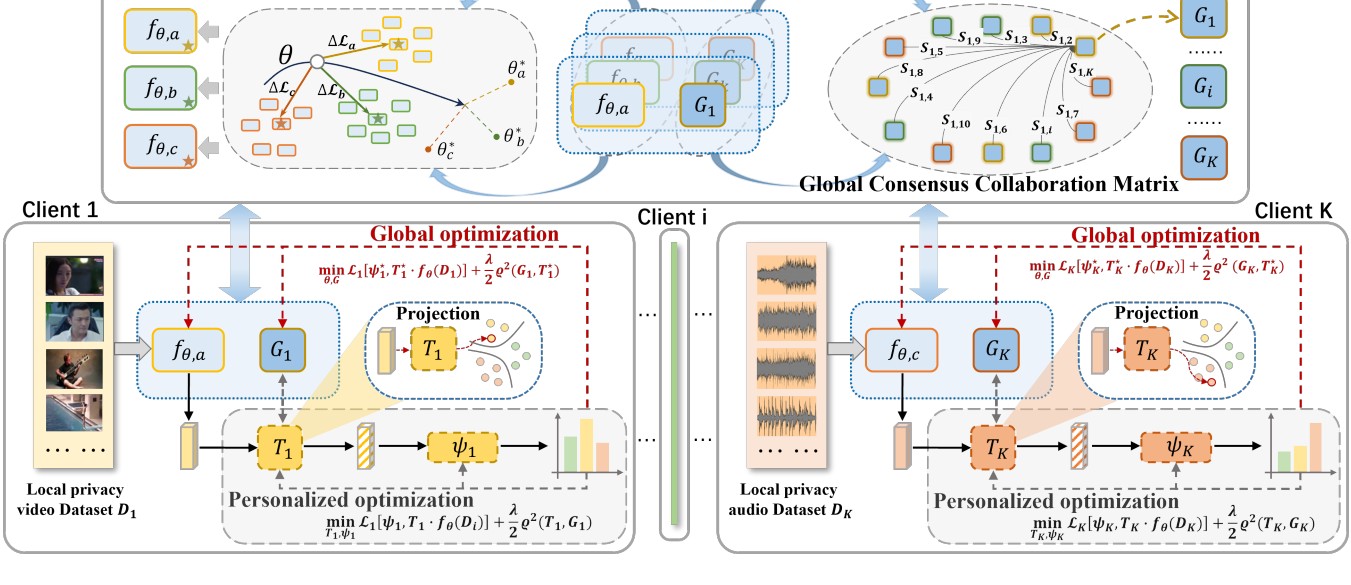

Figure 2: The network architecture of the proposed framework. Client side: Localized Training Via Dual-level Optimization. Server-side: Cross Modal Meta Aggregation. $f_\theta$ is the meta learner, and $G$ is the shared consensus operator.

the final result within the personalized classifier $\psi_i$ associated with $D_i$.

**Client-Side Meta-Knowledge Bi-level Optimization.** Then we present our client-side optimization problem. Given the client data distribution $\{D_i\}_{i=1}^K$, our objective is to learn the base meta learner $f_\theta$ and the shared consensus operator $G$. This enables us to address various client data distributions $D_i$ and discover specific subspace projection personalized operator $T_i$ and tailored feature space multi-class classifiers $\psi_i$ through the following *personalized optimization* learning process:

$$\min_{T_i,\psi_i} \mathcal{L}_i[\psi_i, T_i \cdot f_\theta(D_i)] + \frac{\lambda}{2}\varrho^2(T_i, G_i), \qquad (3)$$

where $T_i$ and $G_i$ are characterized by the property of being orthogonal projections in an $d \times n$ matrix; $\mathcal{L}_i$ is the empirical classification risk (i.e., cross entropy loss function) for the $i$-client data distribution $D_i$; $\psi_i$ is personality classifier.

And then, our approach relies on a dual-layer optimization meta-learning method to learn shared meta learner $f_\theta$ and the shared consensus operator ($G$). This is contingent upon collaborative efforts from all clients for global optimization, facilitating the process of knowledge sharing. The overall *global optimization* objective in federated learning is as follows:

$$\min_{\theta,G} \frac{1}{K}\sum_{i=1}^K \{\mathcal{L}_i[\psi_i^\star, T_i^\star \cdot f_\theta(D_i)] + \frac{\lambda}{2}\varrho^2(G_i, T_i^\star)\}, \qquad (4)$$

where $\psi_i^\star, T_i^\star$ represent the parameters after personalized optimization, and $K$ is the total number of clients.

The objective of the personalized optimization is to precisely extract the optimal personalized subspace projection operator $T$ using shared consensus operators $G$, and to customize personalized multi-class classifiers $\psi$ based on the distribution of client's private

data. Meanwhile, the global optimization aims to propagate the shared knowledge of personalized operators and collaboratively optimize the shared meta learner $f_\theta$ among clients.

## 3.3 Cross Modal Meta Aggregation

**Gradient Consistency-based Clustering.** Due to the heterogeneity of data, particularly in terms of modalities, tasks, and distributions, a singular federated global model struggles to adapt to the model gradient update directions of each client effectively. Previous research [52] highlights that gradient conflicts in meta-learning, especially when dealing with disparate data distributions, impede the model learning process. In such scenarios, the presence of heterogeneous modal data exacerbates gradient conflict issues, consequently impacting training speed. To address this challenge, we propose gradient consistency-based clustering strategy, optimizing the diversity conflicts of different client update directions, thereby significantly enhancing the efficiency of federated communication.

In each communication round, the server receives meta-learner models transmitted by clients. By employing a central averaging algorithm $\bar{\theta} = \frac{1}{K}\sum_{i=1}^K \theta_i$, the server calculates the spatial centroid position $\bar{\theta}$ of these model parameters, enabling the derivation of gradient update directions for each client's meta-learner. Then, we employ the Pearson correlation [15] coefficient of directional data as the measure of directional similarity. The calculation of the similarity between the gradient update directions of models $i$ and $j$ is as follows:

$$\sigma_{ij} = \frac{(\Theta_i - \bar{\Theta}_i) \cdot (\Theta_j - \bar{\Theta}_j)}{\sqrt{(\Theta_i - \bar{\Theta}_i)^2} \times \sqrt{(\Theta_j - \bar{\Theta}_j)^2}}, s.t. \bar{\Theta}_{(i/j)} = \frac{1}{|\Theta|}\sum_{z=1}^{|\Theta|}\Theta_z, \quad (5)$$

where $\Theta_i$ represents the gradient change vector of client $i$ model parameters $\theta_i$ relative to the spatial centroid $\bar{\theta}$.

Eq.5 provides a measure of similarity, denoted as $\sigma$, for the client model update directions. This formula essentially computes the cosine similarity between vectors. Due to the cosine similarity values ranging from -1 to 1 ($\sigma \in [-1, 1]$), although they can be utilized to assess conflicts in gradient directions, they cannot directly serve as indicators of collaboration for optimization directions. To derive collaborative optimization relationships, we can employ the following angular distance formula:

$$A_{ij} = 1 - \left( \frac{cos^{-1}(\sigma_{ij})}{\pi} \right), s.t. A_{ij} \in [0, 1]. \tag{6}$$

Using angular distance $A$, we can perform spectral clustering [41] to obtain clusters $\mu$. Finally, by applying basic average aggregation to the meta-models within each cluster, we obtain a gradient-consistent meta learner. Through this strategy, the shared meta learner clustering is no longer heavily influenced by gradient conflicts. Moreover, it preserves the collaborative nature among clients in federated learning. Upon acquiring aggregated models tailored to specific directions, clients can easily attain local optima for personalized tasks with minimal updates to their own data, ultimately enhancing communication efficiency in federated settings. In Section 3.4, we conducted theoretical analysis and research on this clustering strategy.

**Global Consensus Collaboration Matrix.** Different modalities and tasks indeed exhibit diversity in the feature space, especially with significant disparities between feature spaces of different modalities. Simply averaging shared consensus operators may result in inefficient transfer of spatial knowledge. To better learn a consensus-concordant feature subspace, we propose a Global Consensus Collaboration Matrix. Specifically, after computing the following projection measure of Eq.2, we can obtain the collaborative correlation between different shared consensus operators:

$$S_{ij} = \frac{\rho_j \cdot e^{-\varrho^2(G_i, G_j)}}{\sum_{z=1}^{K} \rho_z \cdot e^{-\varrho^2(G_i, G_z)}}, \tag{7}$$

where $e$ is a constant, and $\rho_j$ represents the proportion of data quantity from client $j$. The purpose of $\rho$ is to provide a reward or penalty mechanism based on the comparison of quantities. Through exponential settings, the distances between different feature subspaces are closer, resulting in larger values for their collaborative relationships.

Subsequently, the server can compute Eq.7 using operators shared across multiple clients to obtain the global consensus collaboration matrix $S \in \mathbb{R}^{K \times K}$ for feature-consistent space. For client $i$, the aggregation formula for collaboration based on subspace metrics is represented as $G_i^{t+1} = \sum_j^K S_{ij} \cdot G_j^t$.

## 3.4 Analysis of Gradient Consistency Theory

In this section, we conduct theoretical analysis on the Gradient Consistency-based Clustering strategy. *Symbols used in section 3.4 are completely separate from the rest of the section.*

The main optimization problem in distributed environments stems from conflicting gradients, where the gradients of different clients are shown to diverge through negative inner products, leading to decreased performance. In previous works, Cao et al. [5] suggest weighting client gradients based on the closeness of their angles for aggregation, potentially excluding those with significant

angular discrepancies, while Liu et al. [30] recommend cropping out conflicting gradients to focus on similar gradient updates. We propose the homo-modal gradient consistency aggregation strategy to solve the conflicts caused by different modal distributions. We tackle this issue from two angles:

First, clients of the same modality generate mutually reinforcing gradient information during the training process. We describe this process using the Gâteaux differentiable, and we make the following assumptions:

**Assumption 1 :** *The client's local loss function f(x) is approximately strongly convex and smooth and bounded by a constant C.*

**Assumption 2 :** $f(x)$ *is uniformly continuous. For any $x_1$ and $x_2$ in its domain, $L > 0$ such that $|f(x_1) - f(x_2)| \leq L|x_1 - x_2|$.*

**Definition 1 (Gâteaux Differentiability) :** $f(x)$ *be a matrix-valued function, and $x$ represents the gradient matrix at a particular point. If there exists a matrix $G$ such that for any direction $V$, we have:*

$$\lim_{t \to 0} \frac{f(X + tV) - f(X) - \langle G, V \rangle}{t} = 0, \tag{8}$$

then $f$ is said to be Gâteaux differentiable at $x$, where $G$ contains detailed information about the variation of the gradient update direction at $x$.

For $f(x)$ Gâteaux differentiable at $x_0$, we have gradient matrices $G_i, G_j$, and $G_k$ for clients $i$, $j$ (same modality), and $k$ (different modality). We use the Frobenius norm to measure the differences between these matrices:$\|G_i - G_j\|_F = \sqrt{\sum_{p=1}^{m} \sum_{q=1}^{n} (G_i)_{pq} - (G_j)_{pq}^2}$ and $\|G_i - G_k\|_F = \sqrt{\sum_{p=1}^{m} \sum_{q=1}^{n} (G_i)_{pq} - (G_k)_{pq}^2}$ , then$\|G_i - G_j\|_F \leq \|G_i - G_k\|_F$ , $G(i) = G(j) + o(g)$, showing that the gradient updates for clients with the same modality differ only by a higher-order infinitesimal.

Second, we suggest that the similarity of gradients affects the model aggregation effect. We can measure the similarity of client gradients by projecting the client's gradient onto the plane normal to another client's gradient[4]. The analysis is as follows:

**Definition 2 :** $\theta_{ij}$ *belongs to $[0, \pi]$. If the gradient updates of client $i$ and client $j$ satisfy $\cos(\theta_{ij}) < 0$, their gradient updates are considered conflicting.*

**Definition 3 (Convergence of Line Search Algorithms) :** $\theta_{ij}$ *be the angle between the gradient update directions of any two clients. If there exists a constant $\gamma$ such that $\theta_{ij} < \frac{\pi}{2} - \gamma$, then the gradient update directions of these two clients are considered to be aggregatable.*

After projecting the gradient update $g_i$ of client $i$ and the gradient update $g_j$ of client $j$ to the normal plane, the cosine similarity is computed according to Eq. (9), and if it satisfies Definition 3, the two similar gradient updates are considered to be similarly oriented can be aggregated .

$$\cos \theta_{ij} = \frac{|g_i||g_j|}{g_i^\top g_j} \geq \cos \left( \frac{\pi}{2} - \gamma \right) = -\sin \gamma \tag{9}$$

According to Yu et al. [52], the aggregation based on the same gradient update direction can resolve gradient conflicts, allowing progress towards the objective function in a faster direction, and enhancing the consistency of the model update direction.

**Table 1: Statistics of Federated Datasets for Simulation.**

| Dataset | Clients | Modality | Feature Size | Classes | Total Instance |
|---|---|---|---|---|---|
| AffectNet | 20 | Image | 1408 | 7 | 283.9K |
| Seed | 6 | EEG | 310 | 5 | 29.1K |
| UCF-101 | 8 | Video | 2048 | 101 | 13.3K |
| Epic-Kitchens | 10 | Audio | 1024 | 97 | 34.0K |
| MEAD | 20 | Video | 2048 | 8 | 217.6K |

## 4 EXPERIMENT

### 4.1 Datasets and Baseline.

**DataSets.** We select five integration datasets with varying modalities and task disparities to build our simulation environment, thereby augmenting the comprehensiveness and diversity of our experimental setup. Datasets **AffectNet** [33], **Seed-V** [31], and **MEAD** [42] represent different categories of sentiment recognition, while datasets **Epic-Kitchens** [16] and **UCF-101** [40] correspond to action recognition in the first-person and third-person perspectives, respectively. For all the above five public datasets, we randomly split local instances on each client into the training and test sets with a ratio of 0.8 : 0.2. Additional details are available in the supplementary materials.

**Multitask Setting.** For five datasets, they encompass four different data modalities and are utilized for conducting four distinct task experiments: *7-class image emotion recognition, 5-class EEG emotion recognition, 101-class third-person video action recognition, 97-class first-person audio action recognition, and 8-class video emotion recognition.* To be noted, the difference of label space within the same task also contributes to the data heterogeneity. In our experiments, unless otherwise specified, we conduct experiments on all five datasets simultaneously, and report the average results of all clients on the same dataset. Additional information regarding the datasets is presented in Tab.1.

**Evaluation Metrics.** Following established federated methods, we employ accuracy as the evaluation metric. To be more specific, we compute the accuracy for each client individually and then average the results across different clients for the same dataset. We repeat the training and testing process 5 times, reporting the average accuracy and standard deviation for each dataset.

**Baseline.** We compare our model with multiple state-of-the-art FL algorithms: (1)**Local**: clients separately train their models without any FL collaboration. (2)**FedAvg**[32]: The clients are partitioned into several mutually exclusive groups, ensuring the sharing of identical modal-task data within each group. (3)**Cross-FedAvg**, in addition to FedAvg, incentivizes federated collaboration among diverse modal task datasets, facilitating the exploration of a shared representation. (4)**Meta-HAR**[25]: This method uses meta-learning algorithm MAML to learn an embedding network for the federated Human activity recognition task. (5)**MaT-FL**[3] is an intuitive clustering-based training baseline to tackle the significant data and task heterogeneities. Each client determines aggregation weights by dynamically inferring its "proximity" to other agents. (6)**MCARN**[49] is a modality-collaborative activity recognition network by collaboratively embedding instances on different local clients into a modality-agnostic feature space and producing modality-specific features that cannot be shared across clients with

different modalities. (7)**FedMSplit**[11] is an AlignPFL method assuming latent space alignment, leveraging multimodal split networks to arbitrarily encourages the information sharing between different groups.

### 4.2 Implementation details.

The overall framework of $M^3Fed$ is implemented with Pytorch[36]. For the compared existing methods, we use the publicly released code. Our model and baselines are all trained with SGD optimizer, where the weight decay is set to 1e-5 and the momentum is set to 0.9. In the dual-layer update optimization of client-side local meta-learning models, the learning rate for personalized optimization is set to 0.01, while for global optimization, it is set to 0.001. Across five datasets, the batch size $B$ is configured to 64, the local iterations $E$ are set to 4, and the communication rounds $C$ are set to 300. In our approach, the shared meta-model consists of a four-layer perceptron with ReLU activation function, where the output dimensionality of data features is set to 512. For the subspace projection operator, a matrix of dimensions 512x384 is utilized to transform features into a 384-dimensional low-rank space. The balancing weight $\lambda$ in Eq.3 and Eq.4 is set to 0.6. The hyperparameter for the number of aggregation clusters based on gradient consistency is set to 4. For the Local baseline, the local epoch count is set to 1000 due to the absence of global communication rounds. Unless explicitly specified, other hyper-parameters of the compared baselines are tuned within the range provided by the authors and the best results are reported.

### 4.3 Comparison and Analysis

**Comparison with State-of-the-Art Methods.** Tab.2 shows the experiment results of average test accuracy across five datasets conducted simultaneously at client-side. Overall, our proposed method outperforms baseline approaches on all datasets, indicating the effectiveness of our framework in mitigating gradient conflicts during the aggregation of disparate modalities and thereby facilitating meta-knowledge sharing across different feature spaces. For instance, on the MEAD dataset, our $M^3Fed$ demonstrates performance improvements of 1.66% and 2.42% compared to FedMSplit under heterogeneous data distribution parameters of 1 and 0.4, respectively. It is noteworthy that our method exhibits competitive results against FedMSplit on the Epic-Kitchens dataset, which is attributed to the dataset's long-tailed distribution, posing challenges in federated collaboration for meta-knowledge sharing. Across all five datasets, our method outperforms meta-learning-based federated learning method, Meta-HAR. Specifically, on the MEAD dataset, our $M^3Fed$ achieves performance enhancements of 6.81% and 7.63% under different heterogeneous distribution conditions. These findings underscore the effectiveness of our Gradient Consistency-based Clustering strategy in mitigating gradient conflicts among diverse modality models, thereby addressing the issue of modality imbalance among clients.

**Impact of Data Heterogeneity.** To further evaluate the effectiveness of our model in alleviating cross-data distribution heterogeneity, we conduct experiments involving two different No-IID distributions ($a = 1$ or 0.4). We adopted the data partitioning method used in previous Federated Learning (FL) articles [28, 53, 54, 56],

**Table 2: Comparative analysis of average test accuracy (%) results across five datasets against state-of-the-art methods. Here are the experimental results of two different data heterogeneity settings ($a = 1$ or $0.4$). When the parameter $a$ is smaller, the data partitioning becomes more heterogeneous.**

| Method | Affectnet | | Seed-V | | UCF-101 | | Epic-Kitchens | | MEAD | |
| --- | --- | --- | --- | --- | --- | --- | --- | --- | --- | --- |
| | $a = 1$ | $a = 0.4$ | $a = 1$ | $a = 0.4$ | $a = 1$ | $a = 0.4$ | $a = 1$ | $a = 0.4$ | $a = 1$ | $a = 0.4$ |
| Local | $53.26_{\pm1.21}$ | $51.82_{\pm1.32}$ | $72.57_{\pm2.41}$ | $71.53_{\pm2.19}$ | $67.58_{\pm2.52}$ | $63.37_{\pm1.91}$ | $39.67_{\pm1.73}$ | $35.89_{\pm2.85}$ | $72.51_{\pm3.51}$ | $70.34_{\pm2.62}$ |
| FedAvg | $57.47_{\pm0.94}$ | $56.24_{\pm0.82}$ | $78.97_{\pm2.16}$ | $76.58_{\pm2.38}$ | $70.38_{\pm2.39}$ | $68.72_{\pm2.73}$ | $44.25_{\pm2.59}$ | $42.68_{\pm1.75}$ | $81.74_{\pm2.28}$ | $80.45_{\pm3.63}$ |
| Cross-FedAvg | $56.43_{\pm1.54}$ | $53.73_{\pm1.37}$ | $75.63_{\pm3.46}$ | $71.51_{\pm4.29}$ | $68.57_{\pm4.38}$ | $66.97_{\pm3.94}$ | $43.38_{\pm3.91}$ | $41.84_{\pm3.27}$ | $77.63_{\pm3.41}$ | $73.78_{\pm4.57}$ |
| Meta-HAR | $61.43_{\pm1.65}$ | $60.17_{\pm1.61}$ | $79.87_{\pm2.63}$ | $77.82_{\pm2.27}$ | $73.79_{\pm2.48}$ | $72.15_{\pm2.67}$ | $46.25_{\pm2.71}$ | $41.35_{\pm1.21}$ | $88.61_{\pm2.16}$ | $86.53_{\pm1.82}$ |
| MaT-FL | $59.62_{\pm1.25}$ | $58.45_{\pm1.92}$ | $80.62_{\pm2.51}$ | $78.46_{\pm1.96}$ | $72.72_{\pm2.11}$ | $70.19_{\pm3.27}$ | $47.32_{\pm1.79}$ | $43.59_{\pm1.41}$ | $87.62_{\pm1.33}$ | $86.31_{\pm1.57}$ |
| MCARN | $61.71_{\pm1.62}$ | $61.31_{\pm1.31}$ | $81.13_{\pm1.67}$ | $79.62_{\pm2.56}$ | $74.79_{\pm2.73}$ | $72.46_{\pm2.78}$ | $49.54_{\pm0.96}$ | $45.68_{\pm1.26}$ | $91.72_{\pm1.10}$ | $90.13_{\pm1.23}$ |
| FedMSplit | $62.54_{\pm1.26}$ | $60.73_{\pm1.49}$ | $81.52_{\pm2.52}$ | $80.47_{\pm1.79}$ | $75.16_{\pm1.88}$ | $71.45_{\pm2.79}$ | $50.18_{\pm1.46}$ | $46.53_{\pm1.83}$ | $93.76_{\pm1.54}$ | $91.74_{\pm1.19}$ |
| **Ours** | $\mathbf{64.21}_{\pm0.96}$ | $\mathbf{62.32}_{\pm1.35}$ | $\mathbf{84.34}_{\pm1.56}$ | $\mathbf{81.62}_{\pm2.28}$ | $\mathbf{76.30}_{\pm1.61}$ | $\mathbf{73.62}_{\pm2.46}$ | $\mathbf{50.82}_{\pm0.98}$ | $\mathbf{47.63}_{\pm1.62}$ | $\mathbf{95.42}_{\pm0.86}$ | $\mathbf{94.16}_{\pm1.27}$ |

**Table 3: Test Accuracy (%) on three datasets with increasing client counts. The number of clients is shown within parentheses following the dataset name.**

| Method | Seed-V(18) | UCF-101(24) | MEAD(60) |
| --- | --- | --- | --- |
| Local | 64.78 | 61.64 | 71.42 |
| FedAvg | 73.43 | 65.46 | 78.63 |
| Cross-FedAvg | 70.56 | 64.26 | 73.39 |
| Meta-HAR | 75.26 | 71.35 | 84.51 |
| MaT-FL | 76.92 | 70.22 | 85.62 |
| MCARN | 79.38 | 72.94 | 87.37 |
| FedMSplit | 78.29 | 73.29 | 88.42 |
| **Ours** | **81.57** | **74.36** | **91.84** |

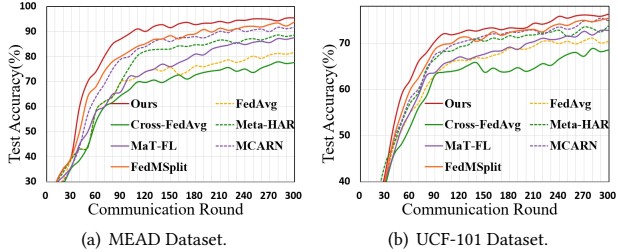

(a) MEAD Dataset.    (b) UCF-101 Dataset.

**Figure 3: Effect of the number of global rounds. (a): The display of average accuracy results for the dataset MEAD at $a = 1$. (b): The display of average accuracy results for the dataset UCF-101 at $a = 1$.**

which involves partitioning data using Dirichlet parameters $a$. Tab.2 shows that in the more heterogeneous partitioning results, our $M^3Fed$ consistently outperforms existing methods, exhibiting significant disparities across various modalities of all datasets. For instance, on the UCF-101 dataset, our $M^3Fed$ achieves a performance improvement of 2.37% compared to FedMSplit.

**Impact of Client number.** As shown in Tab.3, we are conducting testing research with a greater number ($\times3$) of client participants across three datasets. An increased number of client participants will lead to a significant decrease in the volume of training data per client. As expected, our proposed method achieves the best performance across all settings, which further validates that our method can be applied in most practical settings. As the number of clients increases, the performance of all methods decreases and shows some oscillation. However, the performance decrease observed in our approach is minimal. This highlights $M^3Fed$'s ability to facilitate shared meta-knowledge learning more effectively under conditions with a greater number of client participants, thereby enhancing the performance of specific tasks for each client.

**Number of Communication Rounds.** Fig.3 shows the average test accuracy of clients with different number of communication rounds. With a small number of rounds (e.g, less than 90 on the MEAD), our model has similar performance as the baselines, e.g, FedAvg, Cross-FedAvg, and MaT-FL. Due to the proposed low-rank

subspace projection scheme with distinct distribution characteristics, $M^3Fed$ consistently outperforms other baselines in terms of accuracy after undergoing more rounds of training.

### 4.4 Ablation Studies

Here, we present the results for several variants of our model to demonstrate the effectiveness of the primary modules in our $M^3Fed$. To evaluate the performance of the Gradient Consistency-based Clustering module (GCBC), we employ the FedAvg averaging aggregation strategy to replace this module for assessment. Furthermore, to validate the effectiveness of addressing the inconsistency in feature space across different data distributions and the Consensus Subspace Projection Mechanism, we design two elimination studies: eliminating the Global Consensus Collaboration Matrix (GCCM) and eliminating the Consensus Subspace Projection Mechanism (CSPM) by clients.

As shown in Tab.4, the variant model without CSPM performs the worst, suggesting that the transformation through feature low-rank subspace projection facilitates more effective meta-knowledge transfer among clients. This further underscores the role of Consensus Subspace Projection Mechanism in alleviating the disparity issues in feature spaces across different modalities. However, the removal of the Gradient Consistency-based Clustering strategy significantly deteriorates model performance, indicating the aggregated

Table 4: Ablation study of ours method on five datasets.

| Ablation | Affectnet | Seed-V | UCF-101 | EPIC-Kit | MEAD |
|---|---|---|---|---|---|
| $\omega/o$ GCBC | 60.42 | 79.67 | 73.28 | 47.65 | 89.84 |
| $\omega/o$ GCCM | 63.53 | 82.49 | 74.28 | 49.19 | 91.72 |
| $\omega/o$ CSPM | 59.94 | 80.17 | 71.87 | 45.83 | 87.68 |
| **Ours** | **64.21** | **84.34** | **76.30** | **50.82** | **95.42** |

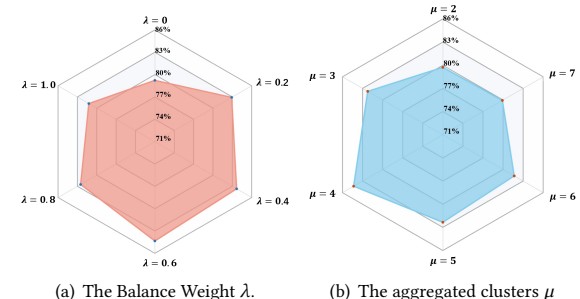

(a) The Balance Weight $\lambda$.      (b) The aggregated clusters $\mu$.

**Figure 4: Effect of the number of hyperparameters. (a): The results on the Seed-V dataset regarding the different balance weights $\lambda$ in Eq.3 and 4. (b): The results on the Seed-V dataset regarding the aggregated clusters $\mu$.**

impact of gradient conflicts in meta-models trained on different modalities. Thus, our aggregation strategy effectively mitigates the challenge of gradient direction conflicts in federated learning. The performance decline is observed when removing the GCCM demonstrates the effectiveness of the proposed Global Consensus Collaboration Matrix aggregation module.

### 4.5 Parametric Analysis

**Analysis of $\lambda$.** In our proposed method, the most critical hyper-parameter is the balancing weight $\lambda$ in the local loss function. Fig.4(a) illustrates the impact of this hyper-parameter on the performance of the Seed-V dataset. As shown, when $\lambda$ is set to 0.6, our $M^3Fed$ achieves optimal performance. However, as the hyper-parameter $\lambda$ approaches 0, the performance sharply declines, indicating the crucial importance of subspace projection mechanism for facilitating meta-knowledge sharing among different modal data. Conversely, when using larger values of $\lambda$, a decrease in performance is observed. This is because an excessive reliance on shared space projection may adversely affect the discriminative ability of individual client classifiers.

**Analysis of Cluster Number $\mu$.** In the Gradient Consistency-based Clustering module, there exists a hyper-parameter that determines the number of clusters $\mu$ for aggregation. As illustrated in Fig.4(b), we conduct experiments on the Seed-V dataset to investigate the impact of this hyper-parameter on performance. In our experiments, when $\mu$ is set to 4, the performance reaches its optimal level. This precisely demonstrates that the aggregation of Gradient Consistency-based Clustering can effectively alleviate the gradient conflict issue in meta-models. However, when $\mu$ takes

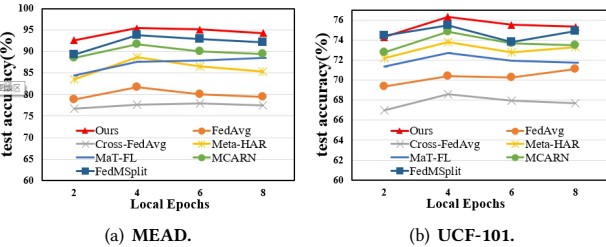

(a) MEAD.      (b) UCF-101.

**Figure 5: Effect of the number of local epochs. (a) and (b) respectively represent the average test results on the MEAD and UCF-101 datasets.**

**Table 5: Average one training run time (s) for clients.**

| Time | Meta-HAR | MCARN | FedMSplit | Ours |
|---|---|---|---|---|
| Seed-V | 3.6s | 3.8s | 5.2s | 3.4s |
| Epic-Kitchens | 6.6s | 7.1s | 8.5s | 6.7s |

other values, the overall performance decreases due to the problem of aggregation of model gradients conflicting among clients.

**Number of Local Epochs $E$.** Fig.5 shows the effect of the number of local updating epochs on the MEAD and the UCF-101 datasets. In our research, we have observed that when the number of local updates reaches 4, our $M^3Fed$ achieves optimal performance, similar to other major baselines. As the number of local updates increases, the meta-model for collaborative training among clients becomes more challenging to achieve globally consistent shared cross-modal meta-models. Conversely, when the number of local updates is small, it leads to slower training speeds, thus failing to achieve optimal accuracy within fewer communication rounds.

**Analysis of Local Train Time.** Tab.5 shows the average train time per epoch on NVIDIA RTX 3090 and Intel(R) Xeon(R) CPU E5-2620. Our $M^3Fed$ outperforms disentangle-based multimodal federated learning on both datasets in terms of time performance. This is because our approach introduces only the optimization computation of the projection operator, thereby reducing the computational load.

## 5 CONCLUSION

In this paper, we propose $M^3Fed$, a meta-learning strategy framework tailored for multi-modal federated learning in a multi-task environment. We introduce a dual-layer meta-learning optimization strategy into multi-modal federated learning to address inter-client modality discrepancies and foster collaboration. To tackle the issue of inconsistent feature spaces across different modalities, we introduce the concept of meta-consensus space to enhance knowledge transfer both within and across modalities. By employing the Gradient Consistency-based Clustering strategy, we address the challenge of federated aggregation caused by disparate data distributions among different modalities. In future work, we will further explore multimodal federated learning in the presence of missing modality data.

Cross-Modal Meta Consensus for Heterogeneous Federated Learning

MM'24, October 28 - November 1, 2024, Melbourne, Australia.

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
