# OpenReview forum: "Cross-Modal Meta Consensus for Heterogeneous Federated Learning"
_acmmm.org/ACMMM/2024/Conference — MM2024 Poster_

### Official Review · Reviewer_WQkR · 2024-05-21

**Rating:** 4
**Confidence:** 4

**Summary:**

The paper proposes a novel Multimodal Federated Learning (MFL) approach called Cross-Modal Meta Consensus (CMMC). This method aims to address the challenges of multimodal data in a federated learning environment, particularly in handling the sharing and integration of information across different modalities and tasks.

**Strengths:**

1.	Innovativeness: The proposed Cross-Modal Meta Consensus method significantly enhances knowledge sharing and task adaptability in multimodal federated learning by introducing new spatial and optimization mechanisms.
2.	Clear Methodology: The paper is well-structured and provides a detailed description of each module's design, including personalized optimization, global optimization, and gradient consistency clustering. It is strongly supported by theory and has substantial practical application potential.

**Limitations:**

1.	Comparison with CreamFL: Please compare your method with CreamFL[1].
2.	Robustness Testing: Please add robustness tests, including data heterogeneity, client quantity, malicious attacks, and sampling rates.
3.	Time Complexity: Please further discuss the time complexity.
[1] Yu Q, Liu Y, Wang Y, et al. Multimodal federated learning via contrastive representation ensemble. ICLR, 2023.

**Suitability:**

3

---

### Official Review · Reviewer_cS6v · 2024-05-23

**Rating:** 4
**Confidence:** 3

**Summary:**

The paper proposes a novel meta-learning strategy for Multimodal Federated Learning (MFL) in a multitask setting, named Cross-Modal Meta Consensus. It aims to harmonize and integrate different modalities (such as text, images, audio) into a unified representation for effective knowledge transfer and task personalization. The method addresses conflicts at meta-initialization points using Gradient Consistency-based Clustering, demonstrating strong performance across diverse data distributions.

**Strengths:**

1. **Innovative Approach**: The introduction of the Cross-Modal Meta Consensus Space to harmonize intra-modal and inter-modal feature spaces is a significant advancement in MFL, facilitating better knowledge transfer and task adaptation.
2. **Theoretical Guarantees**: The paper provides a solid theoretical foundation for the proposed Gradient Consistency-based Clustering, ensuring robust performance and resolving conflicts arising from diverse modality distributions.
3. **Empirical Validation**: The approach is validated on five federated datasets with diverse data distributions, demonstrating its effectiveness in improving performance in multimodal federated learning settings.

**Limitations:**

1. **Complexity**: The dual-level optimization architecture and gradient consistency-based clustering add significant complexity to the model, which may hinder its practical implementation and scalability.
2. **Limited Real-World Applicability**: The evaluation focuses on controlled datasets and may not fully capture the challenges of real-world data heterogeneity and missing modalities.
3. **Computational Overhead**: While the method aims to reduce computational load, the introduction of subspace projection and consensus mechanisms may still result in considerable overhead, particularly for resource-constrained environments.
### Questions for Authors

1. **Scalability**: How does the proposed method scale with an increasing number of clients and modalities? Are there any benchmarks or case studies demonstrating its scalability in large-scale federated learning environments?
2. **Handling Missing Modalities**: How does the model handle scenarios where certain clients have missing modalities? Are there specific strategies or modifications required to maintain performance in such cases?
3. **Real-World Deployment**: What are the practical challenges encountered during the deployment of this method in real-world federated learning systems? Can the authors provide insights or guidelines for practitioners aiming to implement this approach?

**Suitability:**

2

---

### Official Review · Reviewer_bKoz · 2024-05-25

**Rating:** 4
**Confidence:** 3

**Summary:**

The paper presents a significant advancement in the field of multimodal federated learning by integrating meta-learning strategies. The proposed Cross-Modal Meta Consensus and Gradient Consistency-Based Clustering methods are innovative and well-supported by current empirical results. The paper presents an innovative meta-learning strategy designed to address the challenges of multimodal data integration in federated learning and proposes a novel solution to enhance model performance across diverse data distributions.

**Strengths:**

1.The paper introduces a novel concept of Cross-Modal Meta Consensus Space, which is beneficial for knowledge integration and transfer within multimodal data.
2.The authors have not only proposed a new method but also provided theoretical guarantees, including an analysis of gradient consistency-based clustering.
3.The manuscript presents extensive experiments across five different federated datasets, involving a variety of modalities and downstream tasks, which helps to validate the effectiveness of the proposed approach.

**Limitations:**

1.There is a lack of analysis on privacy protection, which is crucial in federated learning.
2.In Sec.3.3, the spatial centroid position is not clear. Whether each local model corresponds to a separate centroid, or all local models together calculate a single centroid, that is what confuses me.
3.The aggregation method is still using just one algorithm, weighted averaging, only the clients are categorized first. This may have very limited novelty to me.
4.The author's writing skills need improvement; there are many parts that are excessively difficult to read, making it hard for readers to comprehend.

**Suitability:**

3

---

### Official Review · Reviewer_4MZx · 2024-05-27

**Rating:** 4
**Confidence:** 3

**Summary:**

This paper propose M3-FL, which utlizes meta-learning with multiltasks on multimodal federated learning. The method are novel and experiments are extensive.

**Strengths:**

1. The method is novel and well-motivated.
2. The experiments are solid and extensive, demonstrating its effectiveness.

**Limitations:**

1. Lack of comparison of some SOTA methods, e.g. CreaMFL (ICLR 23)
2. As of the end of the review date, there is still only one readme file for the public code claimed by the author.

**Suitability:**

3

---

### Meta-Review · Area_Chair_bLRi · 2024-07-02

**Recommendation:** Accept (Poster)
**Confidence:** 5

**Metareview:**

Thank you for your great efforts in the rebuttal. AC recommends this paper to be published.